# Bosworth Fractures of the Ankle: A Systematic Literature Review

**DOI:** 10.3390/jpm13050713

**Published:** 2023-04-23

**Authors:** Ludovico Lucenti, Gianluca Testa, Chiara Nocera, Annalisa Culmone, Eleonora Dell’Agli, Vito Pavone

**Affiliations:** Department of General Surgery and Medical Surgical Specialties, Section of Orthopaedics and Traumatology, University Hospital Policlinico “Rodolico-San Marco”, University of Catania, 95123 Catania, Italy; ludovico.lucenti@gmail.com (L.L.);

**Keywords:** Bosworth, ankle, axilla sign, fracture-dislocation

## Abstract

Bosworth lesions are fracture-dislocations of the ankle and are characterized by entrapment of the proximal segment of the fibula behind the posterior tubercle of the distal tibia. Treatment is challenging, mainly due to failure of a closed reduction. The aim of this study was to review the literature concerning this type of injury. A total of 103 patients with Bosworth fractures were included in the study. The analyzed studies yielded a total of 103 cases, of which 68% (n = 70) were male and 32% (n = 33) were female. Bosworth fractures are mainly due to accidental trauma (58.2%), sports-related injuries (18.4%), and traffic accidents (18.4%). More than 76% of the patients presented a Danis–Weber B fracture, 8.7% a type C fracture, and only 0.97% presented a type A fracture. In 92.2% of the patients, the attempted closed reduction was unsuccessful. A definitive treatment with open reduction and internal fixation (ORIF) was used in 96 patients (93.2%). The most frequent complication was post-traumatic arthritis (10.7%). Bosworth fractures are challenging. The available literature lacks adequate information about this fracture, and an approved standardized algorithm for treating such fractures is not available.

## 1. Introduction

In 1947, Bosworth described for the first time several cases of fracture-dislocation of the ankle, which is characterized by entrapment of the proximal segment of the fibula behind the posterior tubercle of the distal tibia [1].

This lesion is typically produced by supination and external rotation injury patterns [2,3,4,5]. A medial or lateral malleolus fracture may be involved.

A radiographic evaluation is mandatory for diagnosing this type of fracture. In the AP view, an overlap of distal tibia and fibula may be seen, while in the lateral view, a posterior subluxation of the talus and syndesmosis dissociation is usually present. On the CT scan, clearer visualization of the injury is possible, and the fibula can be seen to be displaced behind the posterior edge of the fibular notch (incisura tibiae) and locked between the distal tibia and the displaced posterior malleolus fragment.

Treatment is very challenging and is mainly due to failure of the initial reduction. To treat this complicated injury, open emergency surgery is recommended because a successful closed reduction of Bosworth fracture-dislocation is rare due to soft tissue interposition between the displaced proximal fragment of the fibula and the posterior tubercle of the tibia [6,7].

The aim of this study was to review the most recent literature concerning this type of injury.

## 2. Literature Search

### 2.1. Literature Search Strategy

This systematic review of the current literature was performed according to the Preferred Reporting Items for Systematic Reviews and Meta-Analyses (PRISMA) guidelines [8]. We searched the PubMed, Cochrane Library, and Web of Science databases using the keyword “Bosworth fracture” to identify published case reports or case series evaluating the clinical and radiological characteristics of this fracture, treatment, and short-, medium-, and long-term functional outcomes of patients with Bosworth fractures. The search was restricted to case series, case reports, and prospective and retrospective analyses published in the last 20 years from 1 July 2002 to 1 July 2022. In addition, the reference lists of all possible articles were analyzed to further identify studies potentially suitable for our research, and the articles were evaluated according to the inclusion and exclusion criteria.

### 2.2. Selection Criteria

The studies were selected based on several inclusion criteria: (1) studies written in English language; (2) studies of any level of evidence; (3) studies published after July 1, 2002; (4) studies reporting clinical or preclinical results; and (5) studies concerning the clinical and radiological characteristics, treatment, and functional outcomes in patients with Bosworth fractures. Exclusion criteria consisted of several parameters: (1) inaccessible studies; (2) studies dealing with other topics; and/or (3) studies with poor scientific methodology or without an accessible abstract. Any duplicates were also excluded.

### 2.3. Data Extraction and Criteria Appraisal

Two investigators independently extracted all data from article text, tables, and figures. All data were organized in a specific order: (1) lead author; (2) year of publication; (3) number of patients; (4) participant characteristics; (5) mechanism of injury; (6) classification of fracture; (7) clinical features of the fracture; (8) type and timing of treatment; (9) complications; (10) short-, mid-, and long-term outcomes; (11) time of follow-up; and (12) results. Disagreements between the two reviewers were resolved by consensus. Any remaining controversies were discussed with two senior investigators, who evaluated the quality of the studies. The PRISMA flow chart used for the selection and screening of studies is shown in Figure 1.

## 3. Studies

### 3.1. Study Selection

The initial search of PubMed and Web of Science databases produced 82 results, and 82 articles were examined after duplicate exclusion. Screening of abstracts and titles eliminated 56 articles because they did not meet the inclusion criteria: (1) 31 articles were eliminated from the analysis because the topic was not appropriate, (2) 15 because they were published before 2002, (3) five because they were not in English, and (4) five because they were not available. A total of 26 papers were considered eligible and included in our review. Figure 1 illustrates the study selection process. A total of 103 patients with Bosworth fractures were included in the studies.

### 3.2. Participants

The 26 analyzed studies described sample sizes ranging from a minimum of one patient to a maximum of 51 patients for a total of 103 cases, of which 68% (n = 70) were male, and 32% (n = 33) were female. The mean age was 38.8 years with the youngest patient being 19 years old and the oldest 72 years old. In 47.6% of cases (n = 49), the left was affected, and in 52.4% (n = 54) the right side was (Table 1).

Only the study by Fan et al. [9] reported the body mass indices (BMIs) of patients (BMI = 23 Kg/m^2^), while Won et al. [10] reported the mean body weight (67 kg). Regarding the traumatic mechanism that generated the Bosworth fracture (Table 2), in 58.2% of the patients, the injury was due to accidental trauma (n = 60).

In most of the studies, the Danis–Weber classification was used to characterize the fracture. About three-quarters (76.7%) of the patients (n = 79) presented with a type B fracture, 8.7% (n = 9) with a type C fracture, and only one patient (0.97%) with a type A fracture (Table 3).

Two studies (1.9%) used the Gustilo–Anderson classification because they were open fractures: (1) the case report described by Arora et al. [11] presented a type 2B fracture and (2) the study by Moerenhout et al. [12] described a type 3B fracture. In 11.6% of the patients (n = 12), no data were specified. Radiographically, the “axilla sign” could be seen in these kinds of fractures. This sign was described in only nine patients (8.7%) [4,13,14,15,16,17] while for the rest of the patients, no data were available. A pre-operative computed tomography (CT) scan was performed in 82.5% of patients (n = 85). In four patients (3.9%), the diagnosis was only based on the X-ray study, while in 13.6% (n = 14) the use of X-rays was not reported.

### 3.3. Treatment

From the studies included in our review, it can be seen that a wide variety of treatments have been used for Bosworth fracture (Table 4).

Almost all of the selected studies describe an attempt at closed reduction prior to definitive treatment. In 92.2% of the patients (n = 95), the attempt was unsuccessful, and in only three patients (2.9%) was it successful. Han et al. [18] did not attempt a closed reduction, while Yang et al. [4] did not report the use of this procedure in their study. Surgical treatment was used in all the studies examined, except two in which a non-surgical treatment was chosen with cast for nine weeks [13,19]. Over half (55.3%) of the patients (n = 57) involved in the studies were treated within a day of undergoing trauma, while only 23% (n = 24) were treated after 12 hours. With respect to 12 patients (11.6%), the number of days between injury and surgical treatment was not specified. In total, only five patients (4.8%) were treated with an arthroscopically assisted technique; four patients (3.9%) were treated with plate and screws at the fibula and a syndesmotic screw [20] and one patient (0.97%) with two quadricortical lag screws transversely fixed the syndesmosis [21]. Ninety-six patients (93.2%) were treated with open reduction and internal fixation (ORIF). In 39 patients (37.9%), the ORIF was constructed using a plate and was screwed to the fibula, and in 31 of these (30.1%), a syndesmosis screw was associated. The syndesmosis screws were removed between 12 and 16 weeks in 25 patients, at six weeks in two patients, and at eight weeks in five patients. In two cases, the syndesmosis screw was not removed. The ORIF procedure with screws was only used in four patients (3.9%) [19,22,23]. Moerenhout et al. [12] used K-wires and two partially threaded screws for reducing and fixing the Bosworth fracture in the presented case report. The use of an intramedullary retrograde nail with proximal and distal screws was described by Arora et al. [11]. Won et al. [10] did not specify the type of surgical treatment. After surgical treatment, a cast was placed on 100 patients (97%). Of these, in 18.4% (n = 19), the cast was removed after six weeks, in 14.6% (n = 15) after eight weeks, in only 5.8% (n = 6) was the cast was removed before six weeks, and in 1.9% (n = 2) after two weeks. In 58 patients (56.3%), these data were not specified. Three studies reported use of an external fixator as first intervention for better stability while waiting for swelling and soft tissue edema to subside or because open fractures were present [11,14,23]. Only Cappuccio et al. placed a trans-calcaneal traction to put the limb in traction awaiting definitive surgical treatment [5].

### 3.4. Follow-Up, Post-Operative Complications, and Outcomes

The average follow-up was 15.87 months. Nine patients 8.7% underwent a follow-up of <6 months, 56.3% (n = 58) between 6 and 12 months, 22.3% (n = 23) between 12 and 24 months, and only nine patients (8.7%) more than 24 months (Table 5).

A patient was lost during follow-up, while no data were presented for the remaining three patients (2.9%). Regarding post-operative complications (Table 6), only 34.9% of the patients (n = 36) analyzed in the studies were affected.

Post-traumatic arthritis of the ankle was the most frequently occurring complication in patients (n = 11, 10.7%) [10,14,19] followed by wound complications (n = 8) and mal-union (n = 5) at 7.7% and 4.8%, respectively [10]. Painful joint stiffness in dorsiflexion and plantarflexion was found in six patients (5.8%) [7,19,20]. Joint stiffness and plantarflexed limitation were treated by Cho et al. with open or arthroscopic ankle capsular release [7]. Four patients (3.9%) presented a compartment syndrome followed by fasciotomy the day after surgery [2,10,24]. Bartonicek et al. observed a case of a fixed flexion contracture of the big toe without the possibility of passive correction with ulcerations on the plantar and medial aspect of the great toe and below the first metatarsal head. This toe underwent corrective fusion of the interphalangeal joint of the hallux [24]. Moerenhout et al. described a patient with an invalidating partial necrosis of the talus and pilon one year after surgery that required an ankle–hindfoot arthrodesis 14 months after the accident [12]. A case of shoe wear problems caused by the implant used in the surgical treatment was also described by Schepers et al. [15]. At the end of follow-up, 85.4% of patients (n = 88) who were treated with ORIF showed excellent clinical and radiographic outcomes. The patients presented a good recovery of plantar and dorsal flexion, no lameness, and no functional limitations for the performance of daily life. Two patients (1.94%) treated with ORIF achieved poor results. Among patients treated arthroscopically (4.85%; n = 5), 60% (n = 3) obtained unsuccessful outcomes, and 40% (n = 2) regained ankle function. Instead, cast treatment produced excellent results in one out of two patients. Functional outcomes at the end of follow-up were not reported in six patients (5.8%).

## 4. Discussion

In 1947, David M. Bosworth [1] reported five cases of particular fracture-dislocation of the ankle, in which the proximal fragment of the fibula was incarcerated behind the posterior tibial tubercle. Since that time, this rare ankle fracture pattern has been called “Bosworth fracture” (an example is shown in Figure 2).

According to our analyses, the Bosworth fracture is mostly observed in men with a mean age of 38.8 years old.

Perry et al. performed a cadaveric experiment to identify the trauma mechanism involved in Bosworth fracture [25]. It turned out that the principal mechanism of injury was due to an external rotation force during the supination of the foot (supination-external rotation (SER) fracture according to the Lauge–Hansen classification and associated with Danis–Weber B or C fracture) [15,26]. According to our analysis, a Danis–Weber classification type B was present in 88.76% of the patients, type C in 10.11%, and type A in 1.12%. They described seven stage injury patterns: (1) stage 1: the fibula moves out of the notch after the rupture of the anterior tibiofibular ligament; (2) stage 2: rupture of the posterior tibiofibular ligament; (3) stage 3: rupture of the anteromedial part of the capsule; (4) stage 4: tear of the interosseous membrane; (5) stage 5: the fibula locks posteriorly behind the tibia; (6) stage 6: fracture of the fibula as a result of rotation of the talus; and (7) stage 7: fracture of the medial malleolus or involvement of deltoid ligament. Moerenhout et al. [12] proposed the addition of stage 8 in which a Bosworth fracture is associated with a talus fracture. Regarding the traumatic mechanism from the analysis, it was found that in 58.2%, the fracture was due to an accidental trauma (including falling from a height and falling down stairs), 18.4% due to sports-related injuries, 18.4% due to traffic incidents, 0.97% due to work-related injuries, and 3.9% due to other unspecified causes (Table 1).

In 2007, Bartonicek et al. highlighted age as a factor that can influence the fracture pattern [19]. In a person with skeletal immaturity with open physes, the result of this pattern of fracture is dislocation of the fibula with an epiphysiolysis of the distal tibia. In adults with closed physes, the dislocation of the fibula occurs without fracture. In middle-aged or older people, it is common to see a fibula fracture due to reduced bone elasticity.

The Bosworth fracture represents both a diagnostic and therapeutic challenge for orthopedic surgeons. Usually, in the emergency department, this fracture has a high rate of delayed or missed diagnosis after examination of the initial X-ray images. However, particular characteristics on standard AP, lateral, and mortise radiographs are helpful for making a diagnosis [27]:(1)AP view: overlap of the proximal fibular fragment and the distal tibia;(2)Lateral view: posterior displacement of the fibula and posterior subluxation of the talus;(3)Mortise view: widened medial joint space.

In 2008, Khan and Borton described a new radiographic sign frequently present on an AP mortise view of patients with Bosworth fracture [16]. This sign was defined as the “Axilla sign”, which means cortical radiodensity of the medial tibial plafond occurs because the tibia is locked in internal rotation after dislocation of the fibula occurs. The presence of an Axilla sign should alert the orthopedic surgeon to look for a Bosworth fracture. Moreover, Yang et al. suggested adding an external oblique ankle radiograph to view the position of the fibular shaft compared to the talus [4]. This Axilla sign was described in only 8.7% of all the patients considered in our study.

CT scans and three-dimensional (3D) reconstructions are not routinely used, but they are recommended for ankle fracture-dislocation after failed closed reduction. A CT scan yields additional information about the fracture pattern and interposition of soft tissues, which produces a difficult reduction. Moreover, CT images are useful for pre-operative planning. Our study showed that pre-operative CT scans were obtained in 82.5% of patients.

As mentioned before, treatment of this fracture represents a challenge. Although most ankle fracture-dislocations can be reduced using a closed reduction, in the case of a Bosworth fracture, a successful closed reduction is rare. Closed reduction in this fracture type is usually not possible due to interposition of soft tissues or entrapment of fracture fragments. As Bartonicek’s study reports, a successful closed reduction in Bosworth fractures is rare [6]. In our analysis, the data are similar to Bartobieck as only three out of 103 patients (2.9%) benefitted from a successful closed reduction. Several attempts at performing a closed reduction are not recommended because they can cause further soft tissue damage. Han et al. [18] did not attempt a closed reduction in their case report but rather directly performed an open reduction.

While waiting for surgery, the ankle should be immobilized with a low knee cast and should be elevated. Typically, early open reduction and internal fixation are required to prevent adverse events such as skin necrosis, compartment syndrome [28,29], neurovascular injury, joint stiffness [20], avascular necrosis of the talus [30], and osteoarthritis that can lead to poor functional outcomes.

The ORIF was constructed using a plate and screwing it to the fibula (93.3% of cases). In 30.1% of these cases, a syndesmosis screw was associated. During surgery, it is recommended to test the stability of the syndesmosis after the fixation of fibula. In cases of instability of the syndesmosis, internal fixation is necessary [31]. Lui et al. suggest arthroscopy as a tool to guide the anatomical reduction of syndesmosis and to evaluate associated intra-articular pathology [20]. According to our study, an arthroscopically assisted technique was used in 4.8% of all cases.

In general, the goal of intra-articular fracture management is to achieve anatomical reduction to reduce the risk of developing osteoarthrosis with associated poor functional outcomes [32].

Our study showed that post-traumatic arthritis of the ankle occurred in 10.7% of patients [10,14,19] followed by wound complications (7.7%) and malunion (4.8%) [10]. Painful joint stiffness in dorsiflexion and plantarflexion was found in 5.8% of all cases.

Fournier et al. [33] did not find a correlation between a good reduction and better functional outcomes. Furthermore, in Moerenhout’s study, even if an anatomical reduction and a good alignment of the ankle with hindfoot were obtained, osteoarthritis development was observed [12].

However, anatomical reduction is also necessary to find an optimal position for the arthrodesis a second time.

## 5. Conclusions

Bosworth fractures are lesser known and understood lesions of the ankle and are sometimes misdiagnosed or overlooked. X-ray is usually sufficient to make a diagnosis, but a CT scan is mandatory to fully understand the fracture pattern. This type of injury is challenging. It should be treated without delay and often needs an ORIF, which is characterized by a good recovery, no complications, and a good level of patient satisfaction, to yield better outcomes. The available literature lacks information and only reports a few case reports that are mainly focused on the possible available treatment options and potential complications. Establishment of a nationwide registry addressing this type of fracture could be an optimal instrument to help gain a better understanding of this injury and to design a standard way to treat the lesion. Further studies are necessary to create a clear and validated diagnostic–therapeutic algorithm.

## Figures and Tables

**Figure 1 jpm-13-00713-f001:**
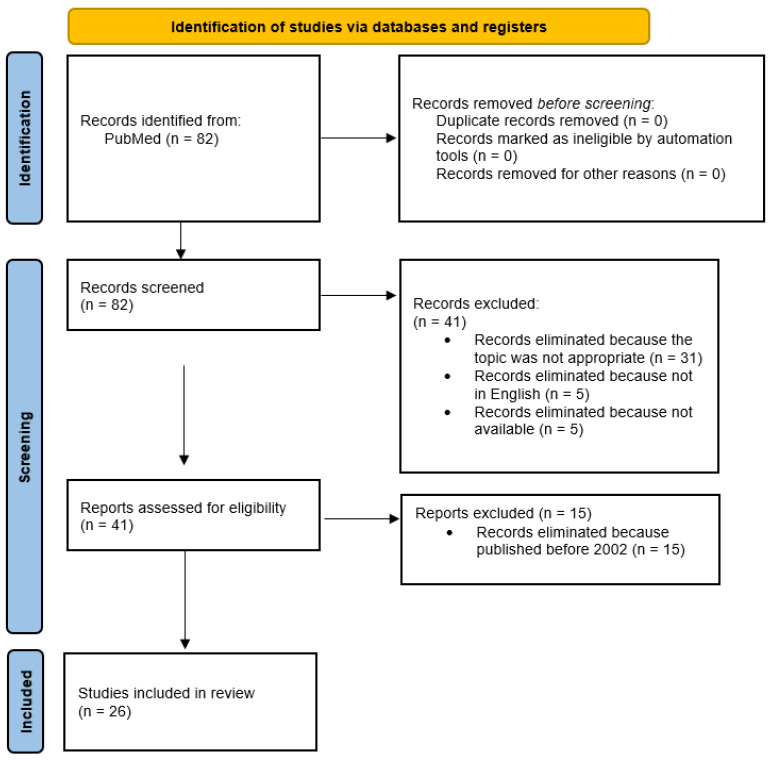
The Preferred Reporting Items for Systematic Reviews and Meta-Analyses (PRISMA) flow chart used for study selection and screening.

**Figure 2 jpm-13-00713-f002:**
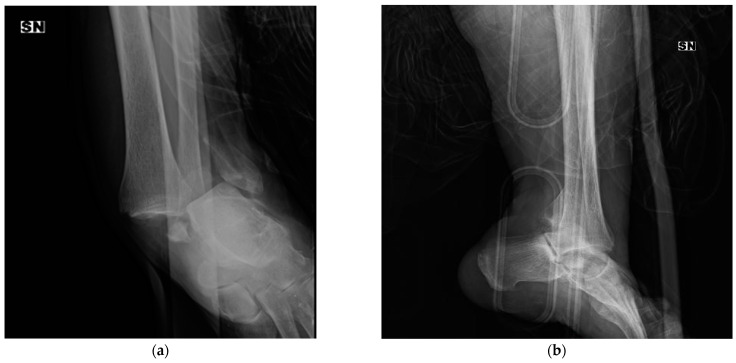
X-ray of a Bosworth fracture (**a**) anterio–posterior (AP) pre-operative view, (**b**) lateral pre-operative view, (**c**,**d**) post-operative X-rays, and (**e**,**f**) the 4-month follow-up.

**Table 1 jpm-13-00713-t001:** Main parameters for each study.

AUTHOR	N° PATIENTS	ARTICLE	SEX
Arora 2021	1	Ankle Arthrodesis Using Retrograde Nail in Case of Bosworth Fracture with Ankle Dislocation: A Rare Case Report	Female
Bartonicek 2007	6	Bosworth-type fibular entrapment injuries of the ankle: the Bosworth lesion. A report of 6 cases and literature review	(I) Female (II) Male (III) Male (IV) Female (V) Female (VI) Male
Bartonicek 2022	1	Bosworth fracture complicated by unrecognized compartment syndrome: a case report and review of the literature	Male
Cappuccio 2017	1	An uncommon case of irreducible ankle fracture-dislocation: the “Bosworth-like” tibio-fibular fracture	Male
Cho 2019	15	Prognostic factors for intermediate-term clinical outcomes following Bosworth fractures of the ankle joint	11 male; 4 female
Delasotta 2013	1	Surgical management of the posterior fibula fracture dislocation: case report	Male
Ellanti 2013	1	Acutely irreducible ankle fracture dislocation: a report of a Bosworth fracture and its management	Male
Fan 2020	1	A Novel Technique for a Successful Closed Reduction of a Bosworth Fracture-Dislocation of the Ankle	Male
Foldager 2018	2	Bosworth fracture dislocation of the ankle: - Two case reports with perioperative illustration	(I) Male (II) Male
Han 2021	1	Bosworth-type fibular entrapment fracture of the ankle without dislocation: a rare case report and a review of the literature	Male
He 2020	1	Ankle arthroscopy-assisted closed reduction in Bosworth fracture dislocation	Female
Ji 2022	1	Case Report: Bosworth Fracture-Dislocation managed by Closed Reduction and Conservative Treatment	Male
Khan 2008	1	A constant radiological sign in Bosworth’s fractures: “the Axilla sign”	Female
Lu 2016	1	A radiological sign (which we are calling the ‘tongues of flame’ sign) in irreducible trimalleolar fractures of the ankle	Female
Lui 2008	4	Ankle stiffness after Bosworth fracture dislocation of the ankle	(I) Male (II)Male (III) Female (IV) Male
Maertin-Somoza 2020	1	Bosworth fracture. An atypical case of irreducible ankle fracture-dislocation	Female
Moerenhout 2019	1	Association of Bosworth, Pilon, and Open Talus Fractures: A Very Unusual Ankle Trauma	Male
Peterson 2015	1	An Unusual Ankle Injury: The Bosworth-Pilon Fracture	Male
Ren 2019	2	Rare variants of Bosworth fracture-dislocation: Bosworth fracture-dislocation with medial malleolus adduction type fracture	2 Male
Schepers 2012	1	An irreducible ankle fracture dislocation: the Bosworth injury	Female
Wenxian 2019	1	Rare Bosworth Fracture-Dislocation Variant of an Irreducible Distal Fibula Dislocation of the Ankle Without Fibula Fracture	Male
Williams 2018	1	Bosworth Dislocation without Associated Fracture	Male
Won 2019	51	Improved functional outcome after early reduction in Bosworth fracture-dislocation	32 male; 19 female
Wright 2012	1	A contemporary approach to the management of a Bosworth injury	Male
Yang 2014	4	Assessment of Bosworth-type fracture by external oblique radiographs	4 Male
Yeoh 2013	1	Bosworth fracture-dislocation of the ankle: a case report	Male

**Table 2 jpm-13-00713-t002:** Mechanism of injury.

Mechanism of Injury	Percentage	N° of Patients
Accidental trauma	58.2%	60
Sport injury	18.4%	19
Traffic accident	18.4%	19
Working injury	0.97%	1
No data	3.9%	4

**Table 3 jpm-13-00713-t003:** Main characteristics of the trauma.

AUTHOR	INJURY TYPE (WEBER)	Axilla Sign	CT SCAN	CLOSED REDUCTION
Arora	Gustilo 2b	No data	No data	Unsuccessful
Bartonicek	Weber B	No data	Yes	Successful (2nd attempt)
Bartonicek	(I) Weber C (II) Weber C (III) Weber C (IV) Weber B (V) Weber B (VI) Weber B	No data	(I) No data (II) No data (III) No data (IV) Yes (V) no data (VI) No data	(I) Unsuccessful (II) Unsuccessful (III) Unsuccessful (III) Unsuccessful (IV) Unsuccessful (V) Unsuccessful (VI) Unsuccessful
Cappuccio	Weber C	No data	Yes	Unsuccessful
Cho	13 Weber B; 2 Weber C	No data	Yes	14 Unsuccessful; 1 successful
Delasotta	Weber B	Yes	Yes	unsuccessful
Ellanti	Weber C	No data	no data	unsuccessful
Fan	No data	No data	No	Successful (2nd attempt)
Foldager	No data	No data	(I) no data (II) no data	(I) Unsuccessful (II) unsuccessful
Han	Weber B	No data	Yes	Not attempted
He	No data	No data	Yes	Unsuccessful
Ji	Weber B	Yes	Yes	Unsuccessful
Khan	No data	Yes	No data	unsuccessful
Lu	Weber B	No data	Yes	unsuccessful
Lui	(I) No data (II)Weber B (III) Weber B (IV) No data	No data	(I) Yes (II) No data (III) No data (IV) no data	(I) Unsuccessful (II) Unsuccessful (III) Unsuccessful
Maertin-Somoza	No data	No data	No	Unsuccessful
Moerenhout	Gustilo 3B	No data	Yes	unsuccessful
Peterson	The initial radiographs showed a bimalleolar ankle fracture dislocation with a comminuted fracture of the tibial plafond	Yes	Yes	unsuccessful
Ren	Weber B	No data	Yes	Unsuccessful
Schepers	Weber B	Yes	No data	unsuccessful
Wenxian	No data	No data	Yes	Unsuccessful
Williams	No data	No data	No	Unsuccessful
Won	1 Weber A; 48 Weber B; 2 Weber C	No data	Yes	Unsuccessful
Wright	No data	No data	Yes	unsuccessful
Yang	4 Weber B	Yes	Yes	
Yeoh	Weber B	No data	No	Unsuccessful

**Table 4 jpm-13-00713-t004:** Main characteristics of the treatments.

AUTHOR	DEFINITIVE TREATMENT	Cast after Surgery	Syndesmotic Screw Removed (weeks/months)	HARDWARE Removed	USE OF EXTERNAL FIXATION	USE OF TRACTION	USE OF CAST
Arora	ORIF	6 weeks			Yes	No	Yes
Bartonicek	ORIF	6 weeks	2 months	Not reported	No	No	Yes
Bartonicek	(I) ORIF (II) cast(III) ORIF (IV) ORIF (V) ORIF (VI) ORIF	(I) 5 weeks (II)–(III) 6 weeks (IV)6 weeks (V) 6 weeks (VI) 6 weeks	(I) 8 weeks after surgery (II)–(III) suprasyndesmotic screws were removed after 2 months (IV) suprasyndesmotic screw was removed 4 months after (V) patient declined removal of the suprasyndesmotic screw (VI) -	(I) Yes, after 2 years (IV) Yes, after 11 month	No	No	Yes
Cappuccio	ORIF	4 weeks			No	Yes	Yes
Cho		8 weeks	12–16 weeks after surgery		No	No	Yes
Delasotta	ORIF	6 weeks	4 months after surgery		No	No	Yes
Ellanti	ORIF	6 weeks	3 months after surgery		No	No	Yes
Fan	ORIF	6 weeks	No data	Not reported	No	No	Yes
Foldager	ORIF	(I) 6 weeks (II) 6 weeks	(I) no data (II) no data		No	No	Yes
Han	ORIF	6 weeks	No syndesmotic screw	Not reported	No	No	Yes
He	Ankle arthroscopy and ORIF	4 weeks	3 months after surgery		No	No	Yes
Ji	cast	9 weeks			No	No	Yes
Khan	ORIF	6 weeks	6 weeks after surgery		No	No	Yes
Lu	ORIF	6 weeks			No	No	Yes
Lui	Arthroscopy and ORIF	(I) 6 weeks (II) no cast (III) no cast (IV) no cast	(I) 12 weeks after surgery (II) 12 weeks after surgery (III)12 weeks after surgery (IV)12 weeks after surgery		No	No	Yes
Maertin-Somoza	ORIF	10 days	No	Not reported	No	No	Yes
Moerenhout	ORIF	12 weeks			No	No	Yes
Peterson	ORIF	6 weeks	No data		Yes	No	Yes
Ren	ORIF	2 weeks	8 weeks	Not reported	No	No	Yes
Schepers	ORIF	6 weeks	Syndesmotic screws were not removed in the absence of complaints	yes, after 9 months (because of shoe wear problems caused by the implant)	No	No	Yes
Wenxian	ORIF	6 weeks	6 weeks after surgery	Yes	Yes	No	Yes
Williams	ORIF	No data	4 months	4 months	No	No	Yes
Won	ORIF	no data	No data	no data	No data	No data	No data
Wright	ORIF	6 weeks			No	No	Yes
Yang	ORIF	No data	No data	Not reported	No data	No data	No data
Yeoh	ORIF	No data	16 weeks	Not reported	No	No	No data

**Table 5 jpm-13-00713-t005:** Complications after Bosworth fractures.

AUTHOR	Complication/Sequelae	Type of Complication	OUTCOME	FOLLOW UP
Arora	No data	No data	No data	No data
Bartonicek	Compartment syndrome	A fixed fexion contracture of the great toe was observed without the possibility of passive correction, with ulcerations on the plantar and medial aspect of the great toe and below the first metatarsal head	Unsuccessful	3 years
Bartonicek	(I) No (II) Yes (III) No (IV) Yes (V) No (VI) No	(I) The patient had no subjective complaints (II) severe posttraumatic osteoarthritis of the ankle and an arthrodesis was performed (III) the patient had no subjective complaints (IV) pain (V) no (VI) no	(I) Successful (II) unsuccessful (III) successful (IV) successful (V) successful (VI) successful	(I) 8 years (II) 2 years (III) 5 years (IV) 2.5 years (V) 3.5 years (VI) 3 years
Cappuccio	No data	No data	Successful	1 year
Cho	2 patients with joint stiffness in dorsiflexion and plantarflexion	Joint stiffness	Successful	2 years
Delasotta	No		Successful	6 months; 8 months
Ellanti	No	No complication	Successful	3 months
Fan	No	No	No data	No data
Foldager	(I) No (II) No	(I) No complication (II) No complication	(I) Successful (II) successful	(I) Lost follow up (II) 3 months
Han	No	No	Successful	2 years
He	No		Successful	6 months
Ji	No	No complication	Successful	9 months; 2.5 years
Khan	No		Successful	3 months
Lu	No		Successful	3 months
Lui	(I) Yes (II) Yes (III) Yes (IV) No	(I) Painful stiffness and using stairs, plantarflexed limitation (open anterior ankle capsular release and extensor tendon adhesiolysis at the level of superior extensor retinaculum was performed. Posterior ankle capsulectomy was also performed through posterior ankle endoscopy) (II) painful stiffness (arthroscopic ankle capsular release was performed) (III) painful stiffness (arthroscopic ankle capsular release was performed)	(I) Unsuccessful (II) unsuccessful (III) unsucessful (IV) successful	(I) 5 months (II) 3 months (III) 3 months (IV) No data
Maertin-Somoza	No	No	Successful	18 months
Moerenhout	No acute complication, but after 1 year he had an invalidating partial necrosis of the talus and pilon	No acute complication, but after 1 year he had an invalidating partial necrosis of the talus and pilon	Unsuccessful	14 months
Peterson	At 18 months, the patient was struggling owing to pain in the ankle that developed clinical and radiologic post-traumatic arthritis	At 18 months, the patient was struggling owing to pain in the ankle that developed clinical and radiologic post-traumatic arthritis	Successful	1.5 years
Ren	No	1 compartment syndrome followed by fasciotomy the day after surgery	Successful	21 months
Schepers	Problems with wearing shoes caused by the implant	Problems eith wearing shoes caused by the implant	Successful	1 years
Wenxian	No	No	Successful	22 months
Williams	No	No	Successful	6 months
Won	Yes	8 wound complication; 2 compartment syndrome; 9 osteoarthritis; 5 mal-union	Successful	6 weeks; 3–6–12 months
Wright	No		Successful	3 months;
Yang	No data	No data	No data	7–9–12–34 months
Yeoh	No	No	Successful	3 months

**Table 6 jpm-13-00713-t006:** Complications after Bosworth fractures.

COMPLICATIONS	Percentage	N of Patients
Post-traumatic arthritis	10.7%	11
Wound complications	7.7%	8
Painful joint stiffness	5.8%	6
Malunion	4.8%	5
Compartment syndrome	3.9%	4
Fixed flexion contracture of great toe	0.97%	1
Necrosis of talus and pilon	0.97%	1
Total	34.9%	36

## Data Availability

Not applicable.

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
