# Peer review of "Bosworth Fractures of the Ankle: A Systematic Literature Review"

_jpm, 2023, doi:10.3390/jpm13050713_

Round 1
Reviewer 1 Report
1. In medical literature, the Bosworth fracture is described as a misdiagnosed and mistreated problem in orthopedics. The authors have noticed this problem well. The methodology is adequate, a significant period of 20 years of research, and a reasonable conclusion has been drawn.
2. The text needs significant revision to be clear and usable for clinicians, and numerous spelling errors should be corrected. The author's name must be revised (Maertin instead of Martin).
3. The first inclusion factor is the same as the first exclusion, and the third inclusion is the same as the second inclusion.
4. The tables could be more straightforward and contain much unnecessary data. It needs to be clarified why the order of authors is arranged in that order. In the first table, keep the author's name and the year of work (same field). The second column that should be observed is the number of patients. The affected side is entirely irrelevant (delete). Gender and BMI do not need to be shown as it is not determined for all patients. Age and prevalence are commented on in the text.
5. The second table is well done, only the mechanism of injury in Bosworth fracture should be written in the extension, and the text from the table should not be repeated in the text below Table 2.
6. Table 3 should be combined with Table 1 to make one.
7. The first two pieces of data are in Table 1, and the following - are Weber classification, axilla sign and CT. It is also desirable to compile tables 4 and 5 to contain the following data: definitive treatment, the time before surgery, cast after surgery, complications, outcome and follow-up. The follow-up column needs to be clarified (let me look too).
8. In the tables describing the entire treatment - just express orifice, arthroscopy and conservative treatment—reduction of operation time within 24 hours and after 24 hours is unnecessary. For the procedure's success, the patient must be cared for in the first 24 hours. Complications after the Bosworth fracture should be added in Table 6.
9. LINE 256 – sentence unclear (limit up to 24h)
10. Conclusion - complete one sentence about the needs of doing CT and define a successful outcome.
11. Abstract - remove lines 13, 11 and 12 - literature searched using keywords.
Author Response
Response to Reviewer 1 Comments
Point 1: In medical literature, the Bosworth fracture is described as a misdiagnosed and mistreated problem in orthopedics. The authors have noticed this problem well. The methodology is adequate, a significant period of 20 years of research, and a reasonable conclusion has been drawn.
Response 1: Thank you very much for the comment. We appreciate.
Point 2: The text needs significant revision to be clear and usable for clinicians, and numerous spelling errors should be corrected. The author's name must be revised (Maertin instead of Martin).
Response 2: Thank you very much for the comment. We changed it.
Point 3. The first inclusion factor is the same as the first exclusion, and the third inclusion is the same as the second inclusion.
Response 3: Thank you very much. We removed the repeated criteria
Point 4. The tables could be more straightforward and contain much unnecessary data. It needs to be clarified why the order of authors is arranged in that order. In the first table, keep the author's name and the year of work (same field). The second column that should be observed is the number of patients. The affected side is entirely irrelevant (delete). Gender and BMI do not need to be shown as it is not determined for all patients. Age and prevalence are commented on in the text.
Response 4: Thank you very much. We removed the columns you requested, we made the changes you seggested and organized in alphabetic order
Point 5. The second table is well done, only the mechanism of injury in Bosworth fracture should be written in the extension, and the text from the table should not be repeated in the text below Table 2.
Response 5: Thank you, we removed it. We kindly ask you to specify what you mean by "written in extension" and we will be glad to change it as you suggest.
Point 6. Table 3 should be combined with Table 1 to make one.
Response 6: Thank you, we totally agree with that. We tried to do it many times, unfortunately, at this point is not possible for us to change it without compromising the style of the table (is too large and it is not possible to see the entire table in a word format). However, we are confident that afterwards this graphic problem may be easily fixed.
Point 7. The first two pieces of data are in Table 1, and the following - are Weber classification, axilla sign and CT. It is also desirable to compile tables 4 and 5 to contain the following data: definitive treatment, the time before surgery, cast after surgery, complications, outcome and follow-up. The follow-up column needs to be clarified (let me look too).
Response 7: Thank you, we totally agree with that. Ideally one large comprehensive table reporting all the data may be the best. As mentioned before, in this phase we are not able to do it without making the table messy and unreadable. We think this graphic issue can be asily addressed.
Point 8. In the tables describing the entire treatment - just express orifice, arthroscopy and conservative treatment—reduction of operation time within 24 hours and after 24 hours is unnecessary. For the procedure's success, the patient must be cared for in the first 24 hours. Complications after the Bosworth fracture should be added in Table 6.
Response 8: Thank you. Complications after Bosworth fractures are already reported
Point 9. LINE 256 – sentence unclear (limit up to 24h)
Response 9: Thank you, the sentence was unnecessary, we deleted it
Point 10. Conclusion - complete one sentence about the needs of doing CT and define a successful outcome.
Response 10: We added two small sentences.
Point 11. Abstract - remove lines 13, 11 and 12 - literature searched using keywords.
Response 11: Thank you, we removed it, as suggested.
English language editing has been conducted
Reviewer 2 Report
Well written report concerning rare clinical entity that helps the clinical practitioner when encountering the injury.
The report could benefit from a drawing of the injury preferably combined with a typical radiograph of a Bosworth ankle fracture. Furthermore, my advice would be to combine the three extensive tables to one table to enhance readability. Lastly, the conclusion would benefit from a more concise advice on how to proceed scientifically concerning this injury, for example by adding the possibility of conducting a nationwide registry on Bosworth ankle injuries as a means of getting more data to optimize treatment in the future.
Author Response
Response to Reviewer 2 Comments
Point 1: Well written report concerning rare clinical entity that helps the clinical practitioner when encountering the injury.
The report could benefit from a drawing of the injury preferably combined with a typical radiograph of a Bosworth ankle fracture. Furthermore, my advice would be to combine the three extensive tables to one table to enhance readability. Lastly, the conclusion would benefit from a more concise advice on how to proceed scientifically concerning this injury, for example by adding the possibility of conducting a nationwide registry on Bosworth ankle injuries as a means of getting more data to optimize treatment in the future.
Response 1:
- Thank you very much for the comments. We appreciate.
- We added some preop and post-op X-rays of a patient.
- Regarding the tables, we totally agree with your suggestion. However, we were not able to do it without making the table messy and unreadable. We are confident this graphic issue can be easily fixed during the final layout.
- Thank you for the last advice, we added it.
- English language editing has been conducted
Reviewer 3 Report
Systematic review of the literature ready for publication, if a typical X-Ray and /or CT Scan will be added, pre- and postoperatively
Author Response
Response to Reviewer 3 Comments
Point 1: Systematic review of the literature ready for publication, if a typical X-Ray and /or CT Scan will be added, pre- and postoperatively
Response 1: Thank you very much for the comments. We appreciate. We added some preop and post-op X-rays of a patient.
Furthermore, English language editing has been conducted
Round 2
Reviewer 1 Report
The authors corrected some of the requested things.
The paper is still full of spelling mistakes. We hope that the editors will correct it.
The table of complications should bear the name of complications after B fracture.
In Table 1, it is unnecessary to write whether the left or right side is affected.
There is no need for Table 3 to include data on the trauma mechanism for each patient or the treatment outcome itself.
In Table 4, describing each type of treatment in detail is unnecessary. It is enough to write down which method was used (ORIF, arthroscopy, etc...).
Author Response
"Please see the attachment."
